# Reconciling emergences: An information-theoretic approach to identify causal emergence in multivariate data

**Fernando E. Rosas** [1,2,3☯*], **Pedro A. M. Mediano** [4☯*], **Henrik J. Jensen** [3,5,6], **Anil K. Seth** [7,8], **Adam B. Barrett** [7,9], **Robin L. Carhart-Harris** [1], **Daniel Bor** [4]

**1** Center for Psychedelic Research, Department of Brain Science, Imperial College London, London SW7 2DD, UK, **2** Data Science Institute, Imperial College London, London SW7 2AZ, UK, **3** Center for Complexity Science, Imperial College London, London SW7 2AZ, UK, **4** Department of Psychology, University of Cambridge, Cambridge CB2 3EB, UK, **5** Department of Mathematics, Imperial College London, London SW7 2AZ, UK, **6** Institute of Innovative Research, Tokyo Institute of Technology, Yokohama 226-8502, Japan, **7** Sackler Centre for Consciousness Science, Department of Informatics, University of Sussex, Brighton BN1 9QJ, UK, **8** CIFAR Program on Brain, Mind, and Consciousness, Toronto M5G 1M1, Canada, **9** The Data Intensive Science Centre, Department of Informatics, University of Sussex, Brighton BN1 9QJ, UK

☯ These authors contributed equally to this work.
\* f.rosas@imperial.ac.uk (FER); pam83@cam.ac.uk (PAM)

**Data Availability Statement:** All relevant data are within the manuscript and its Supporting information files.

## Abstract

The broad concept of *emergence* is instrumental in various of the most challenging open scientific questions—yet, few quantitative theories of what constitutes emergent phenomena have been proposed. This article introduces a formal theory of causal emergence in multivariate systems, which studies the relationship between the dynamics of parts of a system and macroscopic features of interest. Our theory provides a quantitative definition of *downward causation*, and introduces a complementary modality of emergent behaviour—which we refer to as *causal decoupling*. Moreover, the theory allows practical criteria that can be efficiently calculated in large systems, making our framework applicable in a range of scenarios of practical interest. We illustrate our findings in a number of case studies, including Conway's Game of Life, Reynolds' flocking model, and neural activity as measured by electrocorticography.

## Author summary

Many scientific domains exhibit phenomena that seem to be "more than the sum of their parts"; for example, flocks seem to be more than a mere collection of birds, and consciousness seems more than electric impulses between neurons. But what does it mean for a physical system to exhibit emergence? The literature on this topic contains various conflicting approaches, many of which are unable to provide quantitative, falsifiable statements. Having a rigorous, quantitative theory of emergence could allow us to discover the exact conditions that allow a flock to be more than individual birds, and to better understand how the mind emerges from the brain. Here we provide exactly that: a formal theory of what constitutes causal emergence, how to measure it, and what different "types" of

**Funding:** F.R. is supported by the Ad Astra Chandaria foundation. P.M. and D.B. are funded by the 887 Wellcome Trust (grant no. 210920/Z/18/Z). A.K.S. and A.B.B. are grateful to the Dr. Mortimer and Theresa Sackler Foundation, which supports the Sackler Centre for Consciousness Science. The funders had no role in study design, data collection and analysis, decision to publish, or preparation of the manuscript.

**Competing interests:** The authors have declared that no competing interests exist.

emergence exist. To do this, we leverage recent developments in information dynamics—the study of how information flows through and is modified by dynamical systems. As part of this framework, we provide a mathematical definition of causal emergence, and also practical formulae for analysing empirical data. Using these, we are able to confirm emergence in the iconic Conway's Game of Life, in certain flocking patterns, and in representations of motor movements in the monkey's brain.

## Introduction

While most of our representations of the physical world are hierarchical, there is still no agreement on how the co-existing "layers" of this hierarchy interact. On the one hand, *reductionism* claims that all levels can always be explained based on sufficient knowledge of the lowest scale and, consequently—taking an intentionally extreme example—that a sufficiently accurate theory of elementary particles should be able to predict the existence of social phenomena like communism. On the other hand, *emergentism* argues that there can be autonomy between layers, i.e. that some phenomena at macroscopic layers might only be accountable in terms of other macroscopic phenomena. While emergentism might seem to better serve our intuition, it is not entirely clear how a rigorous theory of emergence could be formulated within our modern scientific worldview, which tends to be dominated by reductionist principles.

Emergent phenomena are usually characterised as either strong or weak [1]. *Strong emergence* corresponds to the somewhat paradoxical case of supervenient properties with irreducible causal power [2]; i.e. properties that are fully determined by microscopic levels but can nevertheless exert causal influences that are not entirely accountable from microscopic considerations (the case of strong emergence most commonly argued in the literature is the one of conscious experiences with respect to their corresponding physical substrate [3, 4]). Strong emergence has been as much a cause of wonder as a perennial source of philosophical headaches, being described as "uncomfortably like magic" while accused of being logically inconsistent [2] and sustained on illegitimate metaphysics [5]. *Weak emergence* has been proposed as a more docile alternative to strong emergence, where macroscopic features have irreducible causal power in practice but not in principle. A popular formulation of weak emergence is due to Bedau [5], and corresponds to properties generated by elements at microscopic levels in such complicated ways that they cannot be derived via explanatory shortcuts, but only by exhaustive simulation. While this formulation is usually accepted by the scientific community, it is not well-suited to address mereological questions about emergence in scenarios where parts-whole relationships are the primary interest.

Part of the difficulty in building a deeper understanding of strong emergence is the absence of simple but clear analytical models that can serve the community to guide discussions and mature theories. Efforts have been made to introduce quantitative metrics of weak emergence [6], which enable fine-grained data-driven alternatives to traditional all-or-none classifications. In this vein, an attractive alternative comes from the work on *causal emergence* introduced in Ref. [7] and later developed in Refs. [8, 9], which showed that macroscopic observables can sometimes exhibit more causal power (as understood within the framework of Pearl's *do-calculus* [10]) than microscopic variables. However, this framework relies on strong assumptions that are rarely satisfied in practice, which severely hinders its applicability (this point is further elaborated in Section *Relationship with other quantitative theories of emergence*).

Inspired by Refs. [6, 7], here we introduce a practically useful and philosophically innocent framework to study causal emergence in multivariate data. Building on previous work [11], we take the perspective of an experimentalist who has no prior knowledge of the underlying phenomenon of interest, but has sufficient data of all relevant variables that allows an accurate statistical description of the phenomenon. In this context, we put forward a formal definition of causal emergence that doesn't rely on coarse-graining functions as Ref. [7], but addresses the "paradoxical" properties of strong emergence based on the laws of information flow in multivariate systems.

The main contribution of this work is to enable a rigorous, quantitative definition of *downward causation*, and introduce a novel notion of *causal decoupling* as a complementary modality of causal emergence. Another contribution is to extend the domain of applicability of causal emergence analyses to include cases of observational data, in which case causality ought to be understood in the Granger sense, i.e. as predictive ability [12]. Furthermore, our framework yields practical criteria that can be effectively applied to large systems, bypassing prohibitive estimation issues that severely restrict previous approaches.

The rest of this paper is structured as follows. First, Section *Fundamental intuitions* discusses minimal examples of emergence. Then, Section *A formal theory of causal emergence* presents the core of our theory, and Section *Measuring emergence* discusses practical methods to quantify emergence from experimental data. Our framework is then illustrated on a number of case studies, presented in Section *Case studies*. Finally, the Section *Discussion* concludes the paper with a discussion of some of the implications of our findings.

## Fundamental intuitions

To ground our intuitions, let us introduce minimal examples that embody a few key notions of causally emergent behaviour. Throughout this section, we consider systems composed of $n$ parts described by a binary vector $\boldsymbol{X}_t = (X_t^1, \ldots, X_t^n) \in \{0, 1\}^n$, which undergo Markovian stochastic dynamics following a transition probability $p_{\boldsymbol{X}_{t+1}|\boldsymbol{X}_t}$. For simplicity, we assume that at time $t$ the system is found in an entirely random configuration (i.e. $p_{\boldsymbol{X}_t}(\boldsymbol{x}_t) = 2^{-n}$). From there, we consider three evolution rules.

**Example 1**. *Consider a temporal evolution where the parity of $\boldsymbol{X}_t$ is preserved with probability $\gamma \in (0, 1)$. Mathematically,*

$$p_{\boldsymbol{X}_{t+1}|\boldsymbol{X}_t}(\boldsymbol{x}_{t+1}|\boldsymbol{x}_t) = \begin{cases} \frac{\gamma}{2^{n-1}} & \text{if } \oplus_{j=1}^n x_{t+1}^j = \oplus_{j=1}^n x_t^j, \\ \\ \frac{1-\gamma}{2^{n-1}} & \text{otherwise,} \end{cases}$$

*for all $t \in \mathbb{N}$, where $\oplus_{j=1}^n a_j := 1$ if $\sum_{j=1}^n a_j$ is even and zero otherwise. Put simply: $\boldsymbol{x}_{t+1}$ is a random sample from the set of all strings with the same parity as $\boldsymbol{x}_t$ with probability $\gamma$; and is a sample from the strings with opposite parity with probability $1 - \gamma$.*

*This evolution rule has a number of interesting properties. First, the system has a non-trivial causal structure, since some properties of the future state (its parity) can be predicted from the past state. However, this structure is noticeable* only *at the collective level, as no individual variable has any predictive power over the evolution of itself or any other variable (see* Fig 1*). Furthermore, even the complete past of the system $\boldsymbol{X}_t$ has no predictive power over any individual future $X_{t+1}^j$. This case shows an extreme kind of causal emergence that we call "causal decoupling," in which the parity predicts its own evolution but no element (or subset of elements) predicts the evolution of any other element.*

Causal decoupling          Downward causation

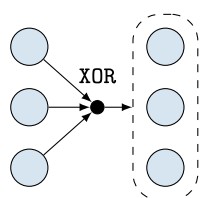        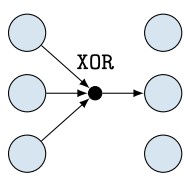

**Fig 1. Minimal examples of causally emergent dynamics.** In Example 1 (*left*) the system's parity tends to be preserved while no interactions occur between low-level elements, which is an example of causal decoupling. In Example 2 (*right*) the system's parity determines one element only, corresponding to downward causation.

**Example 2**. *Consider now a system where the parity of $\boldsymbol{X}_t$ determines $X_{t+1}^1$ (i.e. $X_{t+1}^1 = \oplus_{i=1}^n X_t^i$), and $X_{t+1}^j$ for $j \neq 1$ is a fair coin flip independent of $\boldsymbol{X}_t$ (see* Fig 1*). In this scenario $\boldsymbol{X}_t$ predicts $X_{t+1}^1$ with perfect accuracy, while it can be verified that $X_t^i \perp\!\!\!\perp X_{t+1}^1$ for all $i \in \{1, \ldots, n\}$. Therefore, under this evolution rule the whole system has a causal effect over a particular element, although this effect cannot be attributed to any individual part (for a related discussion, see Ref.* [13]), *being a minimal example of downward causation.*

**Example 3**. *Let us now study an evolution rule that includes the mechanisms of both Examples 1 and 2. Concretely, consider*

$$p_{X_{t+1}|X_t}(\boldsymbol{x}_{t+1}|\boldsymbol{x}_t) = \begin{cases} 0 & \text{if } x_{t+1}^1 \neq \oplus_{j=1}^n x_t^j, \\[2mm] \dfrac{\gamma}{2^{n-2}} & \text{if } x_{t+1}^1 = \oplus_{j=1}^n x_t^j \text{ and } \oplus_{j=1}^n x_{t+1}^j = \oplus_{j=1}^n x_t^j, \\[2mm] \dfrac{1-\gamma}{2^{n-2}} & \text{otherwise.} \end{cases}$$

*As in Example 1, the parity of $\boldsymbol{X}_t$ is transfered to $\boldsymbol{X}_{t+1}$ with probability $\gamma$; additionally, it is guaranteed that $X_{t+1}^1 = \oplus_{i=1}^n X_t^i$. Hence, in this case not only is there a macroscopic effect that cannot be explained from the parts, but at the same time there is another effect going from the whole to one of the parts. Importantly, both effects co-exist independently of each other.*

The above are minimal examples of dynamical laws that cannot be traced from the interactions between their elementary components: Example 1 shows how a collective property can propagate without interacting with its underlying substrate; Example 2 how a collective property can influence the evolution of specific parts; and Example 3 how these two kinds of phenomena take place in the same system. All these issues are formalised by the theory developed in the next section.

## A formal theory of causal emergence

This section presents the main body of our theory of causal emergence. To fix ideas, we consider a scientist measuring a system composed of $n$ parts. The scientist is assumed to measure the system regularly over time, and the results of those measurements are denoted by $\boldsymbol{X}_t = (X_t^1, \ldots, X_t^n)$, with $X_t^i \in \mathcal{X}_i$ corresponding to the state of the $i^{\text{th}}$ part at time $t \in \mathbb{N}$ with phase space $\mathcal{X}_i$. When referring to a collection of parts, we use the notation $\boldsymbol{X}_t^\alpha = (X_t^{i_1}, \ldots, X_t^{i_K})$ for $\alpha = \{i_1, \ldots, i_K\} \subset \{1, \ldots, n\}$. We also use the shorthand notation $[n] := \{1, \ldots, n\}$.

## Supervenience

Our analysis considers two time points of the evolution of the system, denoted as $t$ and $t'$, with $t < t'$. The corresponding dynamics are encoded in the transition probability $p_{X_{t'}|X_t}(\boldsymbol{x}_{t'}|\boldsymbol{x}_t)$. We consider features $V_t \in \mathcal{V}$ generated via a conditional probability $p_{V_t|X_t}$ that are *supervenient* on the underlying system; i.e. that does not provide any predictive power for future states at times $t' > t$ if the complete state of the system at time $t$ is known with perfect precision. We formalise this in the following definition.

**Definition 1**. *A stochastic process $V_t$ is said to be supervenient over $X_t$ if $V_t - X_t - X_{t''}$ form a Markov chain for all $t'' \neq t$.*

The above condition is equivalent to require $V_t$ to be statistically independent of $X_{t''}$ when $X_t$ is given. The relationship between supervenient features and the underlying system is illustrated in Fig 2.

This formalisation of supervenience characterises features $V_t$ that are fully determined by the state of the system at a given time $t$, but also allows the feature to be noisy—which is not critical for our results, but is useful for extending their domain of applicability to practical scenarios. In effect, Definition 1 includes as particular cases deterministic functions $F$ : $\prod_{j=1}^{n} \mathcal{X}_j \rightarrow \mathcal{V}$ such that $V_t = F(X_t)$, as well as features calculated under observational noise— e.g. $V_t = F(X_t) + v_t$, where $v_t$ is independent of $X_t$ for all $t$. In contrast, features that are computed using the values of $X_t$ at multiple timepoints (e.g. the Fourier transform of $X_t$) generally fail to be supervenient.

## Partial information decomposition

Our theory is based on the *Partial Information Decomposition* (PID) framework [14], which provides powerful tools to reason about information in multivariate systems. In a nutshell, PID decomposes the information that $n$ sources $\boldsymbol{X} = (X^1, \ldots, X^n)$ provide about a target variable $Y$ in terms of information atoms as follows:

$$I(\boldsymbol{X}; Y) = \sum_{\boldsymbol{\alpha} \in \mathcal{A}} I_\partial^{\boldsymbol{\alpha}}(\boldsymbol{X}; Y) \ , \tag{1}$$

with $\mathcal{A} = \{\{\alpha_1, \ldots, \alpha_L\} : \alpha_i \subseteq [n], \alpha_i \neg \not\subset \alpha_j \forall i, j\}$ being the set of antichain collections [14]. Intuitively, $I_\partial^{\boldsymbol{\alpha}}$ for $\boldsymbol{\alpha} = \{\alpha_1, \ldots, \alpha_L\}$ represents the information that the collection of variables

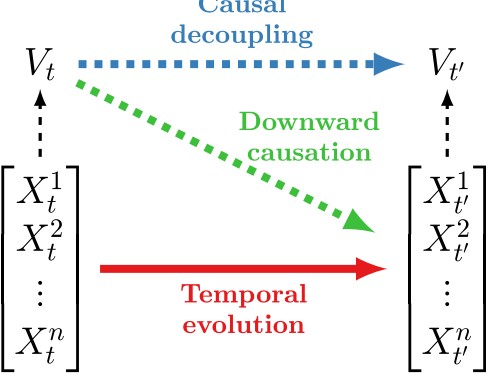

**Fig 2. Diagram of causally emergent relationships.** Causally emergent features have predictive power beyond individual components. Downward causation takes place when that predictive power refers to individual elements; causal decoupling when it refers to itself or other high-order features.

$X^{\alpha_1}, \ldots, X^{\alpha_L}$ provide redundantly, but their sub-collections don't. For example, for $n = 2$ source variables, $\boldsymbol{\alpha} = \{\{1\}\{2\}\}$ corresponds to the information about $Y$ that is provided by both of them, $\boldsymbol{\alpha} = \{\{i\}\}$ to the information provided uniquely by $X_i$, and, most interestingly, $\boldsymbol{\alpha} = \{\{12\}\}$ corresponds to the information provided by both sources jointly but not separately—commonly referred to as *informational synergy*.

One of the drawbacks of PID is that the number of atoms (i.e. the cardinality of $\mathcal{A}$) grows super-exponentially with the number of sources, and hence it is useful to coarse-grain the decomposition according to specific criteria. Here we introduce the notion of $k^{th}$-*order synergy* between $n$ variables, which is calculated as

$$\mathrm{Syn}^{(k)}(\boldsymbol{X}; Y) \sum_{\boldsymbol{\alpha} \in \mathcal{S}^{(k)}} I_\partial^{\boldsymbol{\alpha}}(\boldsymbol{X}; Y) \; ,$$

with $\mathcal{S}^{(k)} = \{\{\alpha_1, \ldots, \alpha_L\} \in \mathcal{A} : \min_j |\alpha_j| > k\}$. Intuitively, $\mathrm{Syn}^{(k)}(\boldsymbol{X};Y)$ corresponds to the information about the target that is provided by the whole $\boldsymbol{X}$ but is not contained in any set of $k$ or less parts when considered separately from the rest. Accordingly, $\mathcal{S}^{(k)}$ only contains collections with groups of more than $k$ sources.

Similarly, we introduce the unique information of $\boldsymbol{X}^\beta$ with $\beta \subset [n]$ with respect to sets of at most $k$ other variables, which is calculated as

$$\mathrm{Un}^{(k)}(\boldsymbol{X}^\beta; Y|\boldsymbol{X}^{-\beta}) \sum_{\boldsymbol{\alpha} \in \mathcal{U}^{(k)}(\beta)} I_\partial^{\boldsymbol{\alpha}}(\boldsymbol{X}; Y) \; .$$

Above, $\mathcal{U}^{(k)}(\beta) = \{\boldsymbol{\alpha} \in \mathcal{A} : \beta \in \boldsymbol{\alpha}, \forall \alpha \neq \beta \in \boldsymbol{\alpha}, \alpha \subseteq [n] \; \beta, |\alpha| > k\}$, and $\boldsymbol{X}^{-\beta}$ being all the variables in $\boldsymbol{X}$ whose indices are not in $\beta$. Put simply, $\mathrm{Un}^{(k)}(\boldsymbol{X}_t^\beta; Y|\boldsymbol{X}^{-\beta})$ represents the information carried by $\boldsymbol{X}^\beta$ about $Y$ that no group of $k$ or less variables within $\boldsymbol{X}^{-\boldsymbol{\beta}}$ has on its own. Note that these coarse-grained terms can be used to build a general decomposition of $I(\boldsymbol{X}, Y)$ described in S1 Appendix (Section 1), the properties of which are proven in S1 Appendix (Section 2).

One peculiarity of PID is that it postulates the structure of information atoms and the relations between them, but it does not prescribe a particular functional form to compute $I_\partial^{\boldsymbol{\alpha}}$. In fact, only one of the information atoms must be specified to determine the whole PID—usually the redundancy between all individual elements [14]. There have been multiple proposals for specific functional forms of $I_\partial^{\boldsymbol{\alpha}}$ in the PID literature; see e.g. Refs. [15–18]. A particular method for fully computing the information atoms based on a recent PID [19] is discussed in Section *Measuring emergence via synergistic channels*.

Conveniently, our theory doesn't rely on a specific functional form of PID, but only on a few basic properties that are precisely formulated in S1 Appendix (Section 2). Therefore, the theory can be instantiated using any PID—as long as those properties are satisfied. Importantly, as shown in Section *Practical criteria for large systems*, the theory allows the derivation of practical metrics that are valid independently of the PID chosen.

## Defining causal emergence

With the tools of PID at hand, now we introduce our formal definition of causal emergence.

**Definition 2**. *For a system described by $\boldsymbol{X}_t$, a supervenient feature $V_t$ is said to exhibit causal emergence of order $k$ if*

$$\mathrm{Un}^{(k)}(V_t; \boldsymbol{X}_{t'}|\boldsymbol{X}_t) > 0 \; . \tag{2}$$

Accordingly, causal emergence takes place when a supervenient feature $V_t$ has irreducible causal power, i.e. when it *exerts causal influence that is not mediated by any of the parts of the system*. In other words, $V_t$ represents some emergent collective property of the system if: 1) contains information that is dynamically relevant (in the sense that it predicts the future evolution of the system); and 2) this information is beyond what is given by the groups of $k$ parts in the system when considered separately.

To better understand the implications of this definition, let us study some of its basic properties.

**Lemma 1**. *Consider a feature $V_t$ that exhibits causal emergence of order 1 over $X_t$. Then,*

1. *The dimensionality of the system satisfies $n \geq 2$.*

2. *There exists no deterministic function $g(\cdot)$ such that $V_t = g(X_t^j)$ for any $j = 1, \ldots, n$.*

*Proof*. See S1 Appendix, Section 3.

These two properties establish causal emergence as a *fundamentally collective* phenomenon. In effect, property (i) states that causal emergence is a property of multivariate systems, and property (ii) that $V_t$ cannot have emergent behaviour if it can be perfectly predicted from a single variable.

In order to use Definition 2, one needs a candidate feature $V_t$ to be tested. However, in some cases there are no obvious candidates for an emergent feature, for which Definition 2 might seem problematic. Our next result provides a criterion for the existence of emergent features based solely on the system's dynamics.

**Theorem 1**. *A system $X_t$ has a causally emergent feature of order $k$ if and only if*

$$\mathtt{Syn}^{(k)}(X_t; X_{t'}) > 0 \ . \tag{3}$$

*Proof*. See S1 Appendix, Section 2.

**Corollary 1**. *The following bound holds for any supervenient feature $V_t$:*
$\mathtt{Un}^{(k)}(V_t; X_{t'}|X_t) \leq \mathtt{Syn}^{(k)}(X_t; X_{t'})$.

This result shows that the capability of exhibiting emergence is closely related to how synergistic the system components are with respect to their future evolution. Importantly, this result enables us to determine whether or not the system admits any emergent features by just inspecting the synergy between its parts—*without knowing what those features might be*. Conversely, this result also allows us to discard the existence of causal emergence by checking a single condition: the lack of dynamical synergy. Furthermore, Corollary 1 implies that the quantity $\mathtt{Syn}^{(k)}(X_t; X_{t'})$ serves as a measure of the *emergence capacity* of the system, as it upper-bounds the unique information of all possible supervenient features.

Theorem 1 establishes a direct link between causal emergence and the system's statistics, avoiding the need for the observer to propose a particular feature of interest. It is important to remark that the emergence capacity of a system depends on the system's partition into microscopic elements—in fact, it is plausible that a system might have emergence capacity under one microscopic representation, but not with respect to another after a change of variables. Therefore, emergence in the context of our theory always refers to "emergence with respect to a given microscopic partition."

## A taxonomy of emergence

Our theory, so far, is able to detect *whether* there is emergence taking place; the next step is to be able to characterise *which kind* of emergence it is. For this purpose, we combine our

feature-agnostic criterion of emergence presented in Theorem 1 with Integrated Information Decomposition, ΦID, a recent extension of PID to multi-target settings [20].

Using ΦID, one can decompose a PID atom as

$$I_\partial^{\boldsymbol{\alpha}}(\boldsymbol{X}_t; \boldsymbol{X}_{t'}) = \sum_{\boldsymbol{\beta} \in \mathcal{A}} I_\partial^{\boldsymbol{\alpha} \to \boldsymbol{\beta}}(\boldsymbol{X}_t; \boldsymbol{X}_{t'}) \ . \tag{4}$$

For example, if $n = 2$ then $I_\partial^{\{1\}\{2\} \to \{1\}\{2\}}$ represents the information shared by both time series at both timesteps (for example, when $X_t^1, X_t^2, X_{t'}^1, X_{t'}^2$ are all copies of each other); and $I_\partial^{\{12\} \to \{1\}}$ corresponds to the synergistic causes in $\boldsymbol{X}_t$ that have a unique effect on $X_{t'}^1$ (for example, when $X_{t'}^1 = X_t^1 \oplus X_t^2$). More details and intuitions on ΦID can be found in Ref. [20].

With the fine-grained decomposition provided by ΦID one can discriminate between different kinds of synergies. In particular, we introduce the *downward causation* and *causal decoupling indices of order k*, denoted by $\mathcal{D}^{(k)}$ and $\mathcal{G}^{(k)}$ respectively, as

$$\mathcal{D}^{(k)}(\boldsymbol{X}_t; \boldsymbol{X}_{t'}) := \sum_{\substack{\boldsymbol{\alpha} \in \mathcal{S}^{(k)} \\ \boldsymbol{\beta} \in \mathcal{A} \, \mathcal{S}^{(k)}}} I_\partial^{\boldsymbol{\alpha} \to \boldsymbol{\beta}}(\boldsymbol{X}_t; \boldsymbol{X}_{t'}) \ , \tag{5}$$

$$\mathcal{G}^{(k)}(\boldsymbol{X}_t; \boldsymbol{X}_{t'}) := \sum_{\boldsymbol{\alpha}, \boldsymbol{\beta} \in \mathcal{S}^{(k)}} I_\partial^{\boldsymbol{\alpha} \to \boldsymbol{\beta}}(\boldsymbol{X}_t; \boldsymbol{X}_{t'}) \ . \tag{6}$$

From these definitions and Eq (4), one can verify that

$$\text{Syn}^{(k)}(\boldsymbol{X}_t; \boldsymbol{X}_{t'}) = \mathcal{G}^{(k)}(\boldsymbol{X}_t; \boldsymbol{X}_{t'}) + \mathcal{D}^{(k)}(\boldsymbol{X}_t; \boldsymbol{X}_{t'}) \ . \tag{7}$$

Therefore, the emergence capacity of a system naturally decomposes in two different components: information about $k$-plets of future variables, and information about future collective properties beyond $k$-plets. The ΦID atoms that belong to these two terms are illustrated within the ΦID lattice for two time series in Fig 3. The rest of this section shows that $\mathcal{D}^{(k)}$ and $\mathcal{G}^{(k)}$ are natural metrics of downward causation and causal decoupling, respectively.

**Downward causation.** Intuitively, downward causation occurs when collective properties have irreducible causal power over individual parts. More formally:

**Definition 3**. *A supervenient feature $V_t$ exhibits downward causation of order k if, for some α with |α| = k*:

$$\text{Un}^{(k)}(V_t; \boldsymbol{X}_{t'}^{\alpha} | \boldsymbol{X}_t) > 0 \ . \tag{8}$$

Note that, in contrast with Definition 2, downward causation requires the feature $V_t$ to have unique predictive power over the evolution of specific subsets of the whole system. In particular, an emergent feature $V_t$ that has predictive power over e.g. $X_{t'}^j$ is said to exert downward causation, as it predicts something about $X_{t'}^j$ that could not be predicted from any particular $X_t^i$ for $i \in [n]$. Put differently, in a system with downward causation the whole has an effect on the parts that cannot be reduced to low-level interactions. A minimal case of this is provided by Example 2 in Section *Fundamental intuitions*.

Our next result formally relates downward causation with the index $\mathcal{D}^{(k)}$ introduced in Eq (5).

**Theorem 2**. *A system $\boldsymbol{X}_t$ admits features that exert downward causation of order k iff $\mathcal{D}^{(k)}(\boldsymbol{X}_t; \boldsymbol{X}_{t'}) > 0$.*

*Proof*. See S1 Appendix, Section 3.

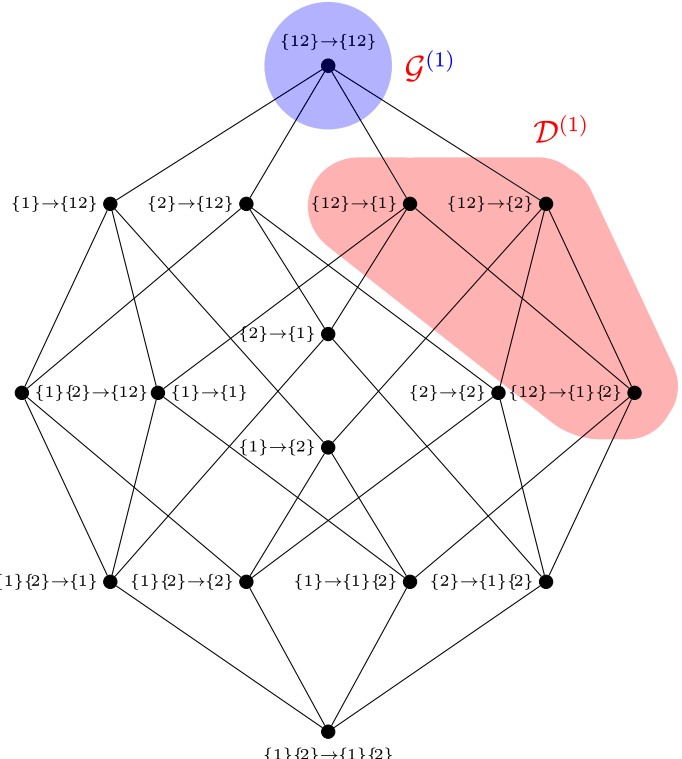

**Fig 3. Integrated information decomposition (ΦID).** ΦID lattice for $n = 2$ time series [20], with downward ($\mathcal{D}$) causation and causal decoupling ($\mathcal{G}$) terms highlighted.

**Causal decoupling.** In addition to downward causation, causal decoupling takes place when collective properties have irreducible causal power over other collective properties. In technical terms:

**Definition 4**. *A supervenient feature $V_t$ is said to exhibit causal decoupling of order k if*

$$\text{Un}^{(k)}(V_t; V_{t'}|\boldsymbol{X}_t, \boldsymbol{X}_{t'}) > 0 \ . \tag{9}$$

*Furthermore, $V_t$ is said to have pure causal decoupling if $\text{Un}^{(k)}(V_t; \boldsymbol{X}_{t'}|\boldsymbol{X}_t) > 0$ and $\text{Un}^{(k)}(V_t; \boldsymbol{X}_{t'}^{\alpha}|\boldsymbol{X}_t) = 0$ for all $\alpha \subset [n]$ with $|\alpha| = k$. Finally, a system is said to be perfectly decoupled if all the emergent features exhibit pure causal decoupling.*

Above, the term $\text{Un}^{(k)}(V_t; V_{t'}|\boldsymbol{X}_t, \boldsymbol{X}_{t'})$ refers to information that $V_t$ and $V_{t'}$ share that cannot be found in any microscopic element, either at time $t$ or $t'$ (note that $\text{Un}^{(k)}(V_t; V_{t'}|\boldsymbol{X}_t, \boldsymbol{X}_{t'})$ is information shared between $V_t$ and $V_{t'}$ that no combination of $k$ or less variables from $\boldsymbol{X}_t$ or $\boldsymbol{X}_{t'}$ has in its own).

Features that exhibit causal decoupling could still exert influence over the evolution of individual elements, while features that exhibit pure decoupling cannot. In effect, the condition $\text{Un}(V_t; X_{t'}^j|\boldsymbol{X}_t) = 0$ implies that the high-order causal effect does not affect any particular part – only the system as a whole. Interestingly, a feature that exhibits pure causal decoupling can be thought of as having "a life of its own;" a sort of *statistical ghost*, that perpetuates itself over time without any individual part of the system influencing or being influenced by it. The system's parity, in the first example of Section *Fundamental intuitions*, constitutes a simple

example of perfect causal decoupling. Importantly, the case studies presented in Section *Case studies* show that causal decoupling can take place not only in toy models but also in diverse scenarios of practical relevance.

We close this section by formally establishing the connection between causal decoupling and the index $\mathcal{G}^{(k)}$ introduced in Eq (6).

**Theorem 3**. *A system possesses features that exhibit causal decoupling if and only if* $\mathcal{G}^{(k)}(\boldsymbol{X}_t; \boldsymbol{X}_{t'}) > 0$. *Additionally, the system is perfectly decoupled if* $\mathcal{G}^{(k)}(\boldsymbol{X}_t; \boldsymbol{X}_{t'}) > 0$ *and* $\mathcal{D}^{(k)}(\boldsymbol{X}_t; \boldsymbol{X}_{t'}) = 0$.

*Proof.* See S1 Appendix, Section 3.

## Measuring emergence

This section explores methods to operationalise the framework presented in the previous section. We discuss two approaches: first, Section *Practical criteria for large systems* introduces sufficiency criteria that are practical for use in large systems; then, Section *Measuring emergence via synergistic channels* illustrates how further considerations can be made if one adopts a specific method of computing ΦID atoms. The latter approach provides accurate discrimination at the cost of being data-intensive and hence only applicable to small systems; the former can be computed in large systems and its results hold independently of the chosen PID, but is vulnerable to misdetections (i.e. false negatives).

### Practical criteria for large systems

While theoretically appealing, our proposed framework suffers from the challenge of estimating joint probability distributions over many random variables, and the computation of the ΦID atoms themselves. As an alternative, we consider approximation techniques that do not require the adoption of any particular PID or ΦID function and are data-efficient, since they are based on pairwise distributions only.

As practical criteria to measure causal emergence of order *k*, we introduce the quantities $\Psi_{t,t'}^{(k)}$, $\Delta_{t,t'}^{(k)}$, and $\Gamma_{t,t'}^{(k)}$. For simplicity, we write here the special case *k* = 1, and provide full formulae for arbitrary *k* and accompanying proofs in S1 Appendix, Section 4:

$$\Psi_{t,t'}^{(1)}(V) := I(V_t; V_{t'}) - \sum_j I(X_t^j; V_{t'}) \ , \tag{10a}$$

$$\Delta_{t,t'}^{(1)}(V) := \max_j \left( I(V_t; X_{t'}^j) - \sum_i I(X_t^i; X_{t'}^j) \right) \ , \tag{10b}$$

$$\Gamma_{t,t'}^{(1)}(V) := \max_j I(V_t; X_{t'}^j) \ . \tag{10c}$$

Our next result links these quantities with the formal definitions in Section *A formal theory of causal emergence*, showing their value as practical criteria to detect causal emergence.

**Proposition 1**. $\Psi_{t,t'}^{(k)}(V) > 0$ *is a sufficient condition for* $V_t$ *to be causally emergent. Similarly,* $\Delta_{t,t'}^{(k)}(V) > 0$ *is a sufficient condition for* $V_t$ *to exhibit downward causation. Finally,* $\Psi_{t,t'}^{(k)}(V) > 0$ *and* $\Gamma_{t,t'}^{(k)}(V) = 0$ *is sufficient for causal decoupling.*

*Proof.* See S1 Appendix, Section 4.

Although calculating whether a system has emergent features via Proposition 1 may be computationally challenging, if one has a candidate feature $V$ one believes may be emergent, one can compute the simple quantities in Eq (10) which depend only on standard mutual information and bivariate marginals, and scales linearly with system size (for $k = 1$). These quantities are easy to compute and test for significance using standard information-theoretic tools [21, 22]. Moreover, the outcome of these measures is valid for any choice of PID and $\Phi$ID that is compatible with the properties specified in S1 Appendix, Section 2.

In a broader context, $\Psi_{t,t'}^{(k)}$ and $\Delta_{t,t'}^{(k)}$ belong to the same *whole-minus-sum* family of measures as the interaction information [14, 23], the redundancy-synergy index [24] and, more recently, the O-information $\Omega$ [25]—which cannot measure synergy by itself, but only the balance between synergy and redundancy. In practice, this means that if there is redundancy in the system it will be harder to detect emergence, since redundancy will drive $\Psi_{t,t'}^{(k)}$ and $\Delta_{t,t'}^{(k)}$ more negative. Furthermore, by summing all marginal mutual informations (e.g. $I(X_t^j; V_{t'})$ in the case of $\Psi_{t,t'}^{(1)}$), these measures effectively *double-count* redundancy up to $n$ times, further penalising the criteria. This problem of double-counting can be avoided if one is willing to commit to a particular PID or $\Phi$ID function, as we show next.

It is worth noticing that the value of $k$ can be tuned to explore emergence with respect to different "scales." For example, $k = 1$ corresponds to emergence with respect to individual microscopic elements, while $k = 2$ refers to emergence with respect to all couples—i.e. individual elements and their pairwise interactions. Accordingly, the criteria in Proposition 1 are, in general, harder to satisfy for larger values of $k$. In addition, from a practical perspective, considering large values of $k$ requires estimating information-theoretic quantities in high-dimensional distributions, which usually requires exponentially larger amounts of data.

## Measuring emergence via synergistic channels

This section leverages recent work on information decomposition reported in Ref. [19], and presents a way of directly measuring the emergence capacity and the indices of downward causation and causal decoupling. The key takeaway of this section is that if one adopts a particular $\Phi$ID, then it is possible to evaluate $\mathcal{D}^{(k)}$ and $\mathcal{G}^{(k)}$ directly, providing a direct route to detect emergence without double-counting redundancy, as the methods introduced in Section *Practical criteria for large systems* do. Moreover, additional properties may become available due to the characteristics of the particular $\Phi$ID chosen.

Let us first introduce the notion of $k$-synergistic channels: mappings $p_{V|X}$ that convey information about $X$ but not about any of the parts $X^\alpha$ for all $|\alpha| = k$. The set of all $k$-synergistic channels is denoted by

$$\mathcal{C}_k(\boldsymbol{X}) = \{p_{V|\boldsymbol{X}} \mid V \perp\!\!\!\perp \boldsymbol{X}^\alpha, \forall \alpha \subseteq [n], |\alpha| = k\}. \tag{11}$$

A variable $V$ generated via a $k$-synergistic channel is said to be a $k$-synergistic observable.

With this definition, we can consider the $k^{\text{th}}$-order synergy to be the maximum information extractable from a $k$-synergistic channel:

$$\mathrm{Syn}_\star^{(k)}(\boldsymbol{X}_t; \boldsymbol{X}_{t'}) := \sup_{\substack{p_{V|\boldsymbol{X}_t} \in \mathcal{C}_k(\boldsymbol{X}_t):\\ V - \boldsymbol{X}_t - \boldsymbol{X}_{t'}}} I(V; \boldsymbol{X}_{t'}) \ . \tag{12}$$

This idea can be naturally extended to the case of causal decoupling by requiring synergistic channels at both sides, i.e.

$$\mathcal{G}_{\star}^{(k)}(\boldsymbol{X}_t; \boldsymbol{X}_{t'}) \coloneqq \sup_{\substack{p_{V|\boldsymbol{X}_t} \in \mathcal{C}_k(\boldsymbol{X}_t), \\ p_{U|\boldsymbol{X}_{t'}} \in \mathcal{C}_k(\boldsymbol{X}_{t'}): \\ V - \boldsymbol{X}_t - \boldsymbol{X}_{t'} - U}} I(V; U) \ . \tag{13}$$

Finally, the downward causation index can be computed from the difference

$$\mathcal{D}_{\star}^{(k)}(\boldsymbol{X}_t; \boldsymbol{X}_{t'}) \coloneqq \mathrm{Syn}_{\star}^{(k)}(\boldsymbol{X}_t; \boldsymbol{X}_{t'}) - \mathcal{G}_{\star}^{(k)}(\boldsymbol{X}_t; \boldsymbol{X}_{t'}) \ . \tag{14}$$

Note that $\mathrm{Syn}_{\star}^{(k)} \geq \mathcal{G}_{\star}^{(k)}$, which is a direct consequence of the data processing inequality applied on $V - \boldsymbol{X}_t - \boldsymbol{X}_{t'} - U$, and therefore $\mathcal{D}_{\star}^{(k)}, \mathcal{G}_{\star}^{(k)} \geq 0$.

By exploiting the properties of this specific way of measuring synergy, one can prove the following result. For this, let us say that a feature $V_t$ is auto-correlated if $I(V_t; V_{t'}) > 0$.

**Proposition 2**. *If $\boldsymbol{X}_t$ is stationary, all auto-correlated k-synergistic observables are $k^{th}$-order emergent.*

*Proof.* See S1 Appendix, Section 4.

In summary, $\mathcal{D}_{\star}^{(k)}$ and $\mathcal{G}_{\star}^{(k)}$ provide data-driven tools to test—and possibly reject—hypotheses about emergence in scenarios of interest. Efficient algorithms to compute these quantities are discussed in Ref. [26]. Although current implementations allow only relatively small systems, this line of thinking shows that future advances in PID might make the computation of emergence indices more scalable, avoiding the limitations of Eq (10).

## Case studies

Let us summarise our results so far. We began by formulating a rigorous definition of emergent features based on PID (Section *Defining causal emergence*), and then used ΦID to break down the emergence capacity into the causal decoupling and downward causation indices (Section *A taxonomy of emergence*). Although these are not straightforward to compute, the ΦID framework allows us to formulate readily computable sufficiency conditions (Section *Practical criteria for large systems*). This section illustrates the usage of those conditions in various case studies. Code to compute all emergence criteria in Eq (10) is provided in an online open-source repository (https://github.com/pmediano/ReconcilingEmergences).

### Canonical examples of putative emergence

Here we present an evaluation of our practical criteria for emergence (Proposition 1) in two well-known systems: Conway's Game of Life (GoL) [27], and Reynolds' flocking boids model [28]. Both are widely regarded as paradigmatic examples of emergent behaviour, and have been thoughtfully studied in the complexity and artificial life literature [29]. Accordingly, we use these models as test cases for our methods. Technical details of the simulations are provided in S1 Appendix, Section 5.

**Conway's Game of Life.**   A well-known feature of GoL is the presence of *particles*: coherent, self-sustaining structures known to be responsible for information transfer and modification [30]. These particles have been the object of extensive study, and detailed taxonomies and classifications exist [29, 31].

To test the emergent properties of particles, we simulate the evolution of 15x15 square cell arrays, which we regard as a binary vector $\boldsymbol{X}_t \in \{0, 1\}^n$ with $n = 225$. As initial condition, we consider configurations that correspond to a "particle collider" setting, with two particles of known type facing each other (Fig 4). In each trial, the system is randomised by changing the

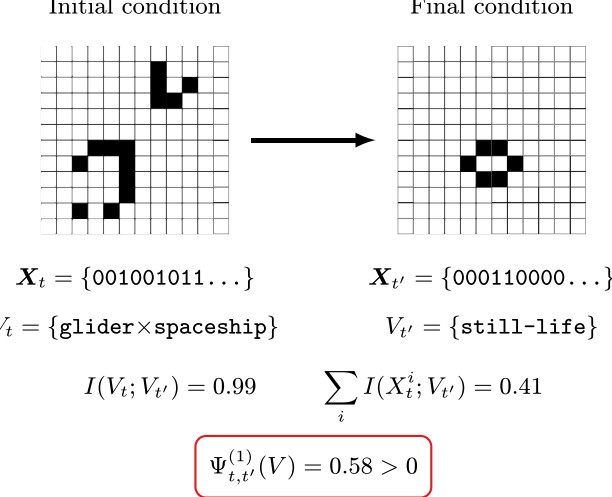

**Fig 4. Causal emergence in Conway's Game of Life.** The system is initialised in a "particle collider" setting, and run until a stable configuration is reached after the collision. Using particle type as a supervenient feature $V$, we find the system meets our practical criterion for causal emergence.

position, type, and relative displacement of the particles. After an intial configuration has been selected, the well-known GoL evolution rule [27] is applied 1000 times, leading to a final state $X_{t'}$. Simulations showed that this interval is enough for the system to settle in a stable state after the collision.

To use the criteria from Eq 10, we need to choose a candidate emergent feature $V_t$. In this case, we consider a symbolic, discrete-valued vector that encodes the type of particle(s) present in the board. Specifically, we consider $\boldsymbol{V}_t = (V_t^1, \ldots, V_t^L)$, where $V_t^j = 1$ iff there is a particle of type $j$ at time $t$—regardless of its position or orientation.

With these variables, we compute the quantities in Eq 10 using Bayesian estimators of mutual information [32]. The result is that, as expected, the criterion for causal emergence is met with $\Psi_{t,t'}^{(1)}(V) = 0.58 \pm 0.02$. Furthermore, we found that $\Gamma_{t,t'}^{(1)}(V) = 0.009 \pm 0.0002$, which is orders of magnitude smaller than $I(V_t; V_{t'}) = 0.99 \pm 0.02$. Errors represent the standard deviation over surrogate data, as described in S1 Appendix, Section 5. Using Proposition 1, these two results suggest that particle dynamics in GoL may not only be emergent, but causally decoupled with respect to their substrate.

**Reynolds' flocking model.** As a second test case, we consider Reynolds' model of flocking behaviour. This model is composed by *boids* (bird-oid objects), with each boid represented by three numbers: its position in 2D space and its heading angle. As candidate feature for emergence, we use the 2D coordinates of the center of mass of the flock, following Seth [6].

In this model boids interact with one another following three rules, each regulated by a scalar parameter [6]:

- **aggregation** ($a_1$), as they fly towards the center of the flock;

- **avoidance** ($a_2$), as they fly away from their closest neighbour; and

- **alignment** ($a_3$), as they align their flight direction to that of their neighbours.

Following Ref. [6], we study small flocks of $N = 10$ boids with different parameter settings to showcase some properties of our practical criterion of emergence. Note that this study is meant as an illustration of the proposed theory, and not as a thorough exploration of the

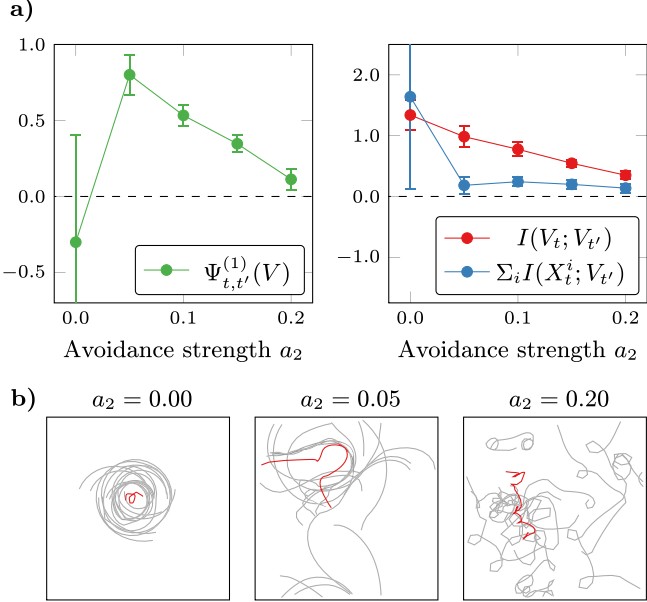

**Fig 5. Causal emergence in the flocking boids model.** As the avoidance parameter is increased, the flock transitions from an attractive regime (in which all boids orbit regularly around a stable center of mass), to a repulsive one (in which boids spread across space and no flocking is visible). **a)** Our criterion $\Psi$ detects causal emergence in an intermediate range of the avoidance parameter (error bars represent the standard deviation estimated over surrogate data). **b)** Sample trajectories of boids (grey) and their center of mass (red).

flocking model, for which a vast literature exists (see e.g. the work of Vicsek [33] and references therein).

Fig 5 shows the results of a parameter sweep over the avoidance parameter, $a_2$, while keeping $a_1$ and $a_3$ fixed. When there is no avoidance, boids orbit around a slowly-moving center of mass, in what could be called an ordered regime. Conversely, for high values of $a_2$ neighbour repulsion is too strong for lasting flocks to form, and isolated boids spread across the space avoiding one another. For intermediate values, the center of mass traces a smooth trajectory, as flocks form and disintegrate. In line with the findings of Seth [6], our criterion indicates that the flock exhibits causally emergent behaviour in this intermediate range.

By studying separately the two terms that make up $\Psi$ we found that the criterion of emergence fails for both low and high $a_2$, but for different reasons (see Fig 5). In effect, for high $a_2$ the self-predictability of the center of mass (i.e. $I(V_t;V_{t'})$) is low; while for low $a_2$ it is high, yet lower than the mutual information from individual boids (i.e. $\sum_i I(X_t^i; V_{t'})$). These results suggest that the low-avoidance scenario is dominated not by a reduction in synergy, but by an increase in redundancy, which effectively increases the synergy threshold needed to detect emergence. However, note that, due to the limitations of the criterion, the fact that $\Psi_{t,t'}^{(1)} < 0$ is inconclusive and does not rule out the possibility of emergence. This is a common limitation of whole-minus-sum estimators like $\Psi$; further refinements may provide bounds that are less susceptible to these issues and perform accurately in these scenarios.

## Mind from matter: Emergence, behaviour, and neural dynamics

A tantalising outcome of having a formal theory of emergence is the capability of bringing a quantitative angle to the archetype of emergence: the mind-matter relationship [35, 36]. As a

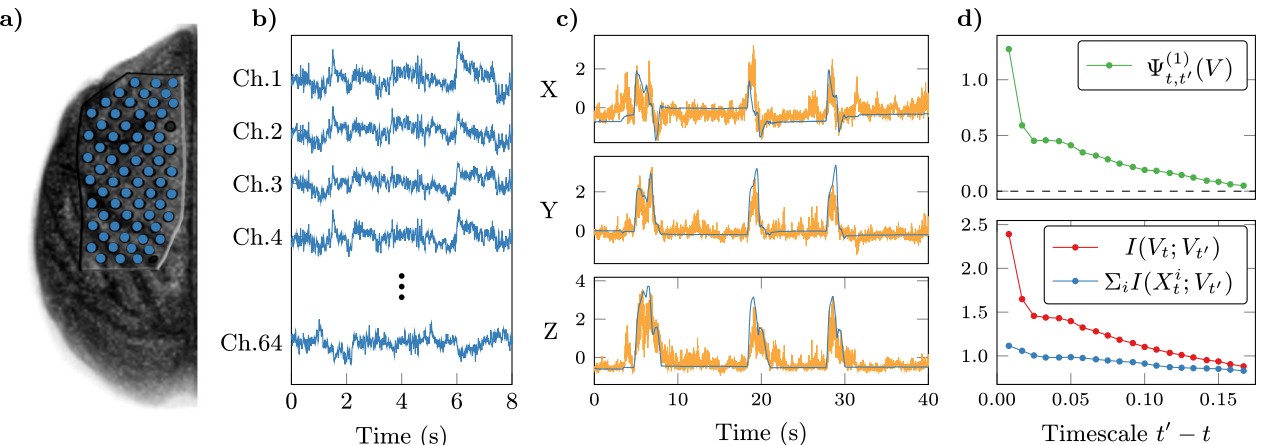

**Fig 6. Causal emergence in motor behaviour of an awake macaque monkey. a)** Position of electrocorticogram (ECoG) electrodes used in the recording (in blue) overlaid on an image of the macaque's left hemisphere (front of the brain towards the top of the page). **b)** Sample time series from the 64-channel ECoG recordings used, which correspond to the system of interest $X_t \in \mathbb{R}^{64}$. **c)** 3D position of the macaques's wrist, as measured by motion capture (blue) and as predicted by the regression model (orange), taken as a supervenient feature $V_t \in \mathbb{R}^3$. **d)** Our emergence criterion yields $\Psi_{t,t'}^{(1)}(V) > 0$, detecting causal emergence of the behaviour with respect to the ECoG sources. Original data and image from Ref. [34] and the Neurotycho database.

first step in this direction, we conclude this section with an application of our emergence criteria to neurophysiological data.

We study simultaneous electrocorticogram (ECoG) and motion capture (MoCap) data of Japanese macaques performing a reaching task [34], obtained from the online Neurotycho database. Note that the MoCap data cannot be assumed to be a supervenient feature of the available ECoG data, since it doesn't satisfy the conditional independence conditions required by our definition of supervenience (see Section *A formal theory of causal emergence*). This is likely to be the case, because the neural system is only partially observed—i.e. the ECoG does not capture every source of relevant activity in the macaque's cortex. Note that non-supervenient features are of limited interest within our framework, as they can satisfy Proposition 1 in trivial ways (e.g. time series which are independent of the underlying system satisfy $\Psi > 0$ if they are auto-correlated). Instead, we focus on the portion of neural activity encoded in the ECoG signal that is relevant to predict the macaque's behaviour, and conjecture this information to be an emergent feature of the underlying neural activity (Fig 6).

To test this hypothesis, we take the neural activity (as measured by 64 ECoG channels distributed across the left hemisphere) to be the system of interest, and consider a memoryless predictor of the 3D coordinates of the macaque's right wrist based on the ECoG signal. Therefore, in this scenario $X_t \in \mathbb{R}^{64}$ and $V_t = F(X_t) \in \mathbb{R}^3$. To build $V_t$, we used Partial Least Squares (PLS) and a Support Vector Machine (SVM) regressor, the details of which can be found in S1 Appendix, Section 6.

After training the decoder and evaluating on a held-out test set, results show that $\Psi > 0$, confirming our conjecture that the motor-related information is an emergent feature of the macaque's cortical activity. For short timescales ($t' - t = 8$ ms), we find $\Gamma_{t,t'}^{(1)}(V) = 0.049 \pm 0.002$, which is orders of magnitude smaller than $\Psi_{t,t'}^{(1)}(V) = 1.275 \pm 0.002$, suggesting that the behaviour may have an important component decoupled from individual ECoG channels (errors are standard deviations estimated over time-shuffled data). Furthermore, the emergence criterion is met for multiple timescales

$t' - t$ of up to $\approx 0.2$s, beyond which the predictive power in $V_t$ and individual electrodes decrease and become nearly identical.

As a control, we performed a surrogate data test to confirm the results in Fig 6 were not driven by the autocorrelation in the ECoG time series. To this end, we re-run the analysis (including training and testing the PLS-SVM) using the same ECoG data, but time-shuffling the wrist position—resulting in a $V_t$ that does not extract any meaningful information from the ECoG, but has the same properties induced by autocorrelation, filtering and regularisation. As expected, the resulting surrogate $\Psi_{t,t'}^{(1)}$ is significantly lower than the one using the un-shuf-fled wrist position, confirming the measured $\Psi_{t,t'}^{(1)}$ is positive and higher than what would be expected from a similar, random projection of the ECoG (details in S1 Appendix, Section 6).

This analysis, while just a proof of concept, helps us quantify how and to what extent behaviour emerges from collective neural activity; and opens the door to further tests and quantitative empirical explorations of the mind-matter relationship.

## Discussion

A large fraction of the modern scientific literature considers strong emergence to be impossible or ill-defined. This judgement is not fully unfounded: a property that is simultaneously supervenient (i.e. that can be computed from the state of the system) and that has irreducible causal power (i.e. that "tells us something" that the parts don't) can indeed seem to be an oxymoron [5]. Nonetheless, by linking supervenience to static relationships and causal power to dynamical properties, our framework shows that these two phenomena are perfectly compatible within the—admittedly counterintuitive – laws of multivariate information dynamics [20], providing a tentative solution to this paradox.

Our theory of causal emergence is about predictive power, not "explicability" [3], and therefore is not related to views on strong emergence such as Chalmers' [3]. Nevertheless, our framework embraces aspects that are commonly associated with strong emergence—such as downward causation—and renders them quantifiable. Our framework also does not satisfy conventional definitions of weak emergence (systems studied in Section *Fundamental intuitions* are not weakly emergent in the sense of Bedau [5], being simple and susceptible to explanatory shortcuts) but is compatible with more general notions of weak emergence, e.g. the one introduced by Seth (see Section *Relationship with other quantitative theories of emergence*). Hence, our theory can be seen as an attempt at reconciling these approaches [36], showing how "strong" a "weak" framework can be.

An important consequence of our theory is the fundamental connection established between causal emergence and statistical synergy: the system's capacity to host emergent features was found to be determined by how synergistic its elements are with respect to their future evolution. Although previous ideas about synergy have been loosely linked to emergence in the past [37], this is (to the best of our knowledge) the first time such ideas have been formally laid out and quantified using recent advances in multivariate information theory.

Next, we examine a few caveats regarding the applicability of the proposed theory, its relation with prior work, and some open problems.

### Scope of the theory

Our theory focuses on *synchronic* [38] aspects of emergence, analysing the interactions between the elements of dynamical systems and collective properties of them as they jointly evolve over time. As such, our theory directly applies to any system with well-defined dynamics, including systems described by deterministic dynamical systems with random initial

conditions [11] and stochastic systems described by Fokker-Planck equations [39]. In contrast, the application of our theory to systems in thermodynamic equilibrium may not be straightforward, as their dynamics are often not uniquely specified by the corresponding Gibbs distributions (for an explicit example, when considering the Ising model, Kawasaki and Glauber dynamics are known to behave differently even when the system is in equilibrium [40]; and thus may provide quite different values of the measures described in Section *Measuring emergence*). Finding principled approaches to guide the application of our theory to those cases is an interesting challenge for future studies.

In addition, given the breadth of the concept of "emergence," there are a number of other theories leaning more towards philosophy that are orthogonal to our framework. This includes, for example, theories of emergence as *radical novelty* (in the sense of features not previously observed in the system) [41], most prominently encapsulated in the aphorism "more is different" by Anderson—see Refs. [42, 43], particularly his approach to emergence in biology (note that some of Anderson's views, particularly the ones related to rigidity, are nevertheless closely related to the approach developed by our framework), and also articulated in the work of Kauffman [44, 45]. Also, *contextual emergence* emphasises a role for macro-level contexts that cannot be described at the micro-level, but which impose constraints on the micro-level for the emergence of the macro [46, 47]. These are valuable philosophical positions, which have been studied from a statistical mechanics perspective in Ref. [46, 48]. Future work shall attempt to unify these other approaches with our proposed framework.

## Causality

The *de facto* way to assess the causal structure of a system is to analyse its response to controlled interventions or to build intervention models (causal graphs) based on expert knowledge, which leads to the well-known *do-calculus* spearheaded by Judea Pearl [10]. This approach is, unfortunately, not applicable in many scenarios of interest, as interventions may incur prohibitive costs or even be impossible, and expert knowledge may not be available. These scenarios can still be assessed via the Wiener-Granger theory of *statistical causation*, which studies the blueprint of predictive power across the system of interest by accounting non-mediated correlations between past and future events [12]. Both frameworks provide similar results when all the relevant variables have been measured, but can neverthelss differ radically when there are unobserved interacting variables [10]. The debate between the Wiener-Granger and the Pearl schools has been discussed in other related contexts—see e.g. Refs. [49, 50] for a discussion regarding Integrated Information Theory (IIT), and Ref. [51] for a discussion about effective and functional connectivity in the context of neuroimaging time series analysis (in a nutshell, effective connectivity aims to uncover the minimal physical causal mechanism underlying the observed data, while functional connectivity describes directed or undirected statistical dependences [51]).

In our theory, the main object of analysis is Shannon's mutual information, $I(X_t; X_{t'})$, which depends on the joint probability distribution $p_{X_t, X_{t'}}$. The origin of this distribution (whether it was obtained by passive observation or by active intervention) will change the interpretation of the quantities presented above, and will speak differently to the Pearl and the Wiener-Granger schools of thought; some of the implications of these differences are addressed when discussing Ref. [7] below. Nonetheless, since both methods of obtaining $p_{X_t, X_{t'}}$ allow synergy to take place, our results are in principle applicable in both frameworks—which allows us to formulate our theory of causal emergence without taking a rigid stance on a theory of causality itself.

### Relationship with other quantitative theories of emergence

This work is part of a broader movement towards formalising theories of complexity through information theory. In particular, our framework is most directly inspired by the work of Seth [6] and Hoel *et al.* [7], and also related to recent work by Chang *et al.* [52]. This section gives a brief account of these theories, and discusses how they differ from our proposal.

Seth [6] proposes that a process $V_t$ is Granger-emergent (or *G-emergent*) with respect to $\mathbf{X}_t$ if two conditions are met: (i) $V_t$ is autonomous with respect to $\mathbf{X}_t$ (i.e. $I(V_t; V_{t'}|\mathbf{X}_t) > 0$), and (ii) $V_t$ is G-caused by $\mathbf{X}_t$ (i.e. $I(\mathbf{X}_t; V_{t'}|V_t) > 0$). The latter condition is employed to guarantee a relationship between $\mathbf{X}_t$ and $V_t$; in our framework an equivalent role is taken by the requirement of supervenience. The condition of autonomy is certainly related with our notion of causal decoupling. However, as shown in Ref. [14], the conditional mutual information conflates unique and synergistic information, which can give rise to undesirable situations: for example, it could be that $I(V_t; V_{t'}|\mathbf{X}_t) > 0$ while, at the same time, $I(V; V_{t'}) = 0$, meaning that the dynamics of the feature $V_t$ are only visible when considering it together with the full system $\mathbf{X}_t$, but not on its own. Our framework avoids this problem by refining this criterion via PID, and uses only the unique information for the definition of emergence.

Our work is also strongly influenced by the framework put forward by Hoel and colleagues in Ref. [7]. Their approach is based on a coarse-graining function $F(\cdot)$ relating a feature of interest to the system, $V_t = F(\mathbf{X}_t)$, which is a particular case of our more general definition of supervenience. Emergence is declared when the dependency between $V_t$ and $V_{t'}$ is "stronger" than the one between $\mathbf{X}_t$ and $\mathbf{X}_{t'}$. Note that $V_t - \mathbf{X}_t - \mathbf{X}_{t'} - V_{t'}$ is a Markov chain, and hence $I(V_t; V_{t'}) \leq I(\mathbf{X}_t; \mathbf{X}_{t'})$ due to the data processing inequality; therefore, a direct usage of Shannon's mutual information would make the above criterion impossible to fulfill. Instead, this framework focuses on the transition probabilities $p_{V_{t'}|V_t}$ and $p_{\mathbf{X}_{t'}|\mathbf{X}_t}$, and hence the mutual information terms are computed using maximum entropy distributions instead of the stationary marginals. By doing this, Hoel *et al.* account not for what the system *actually does*, but for all the potential transitions the system *could do*. However, in our view this approach is not well-suited to assess dynamical systems, as it might account for transitions that are never actually explored. The difference between stationary and maximum entropy distributions can be particularly dramatic in non-ergodic systems with multiple attractors—for a related discussion in the context of Integrated Information Theory, see Ref. [50]. Additionally, this framework relies on having exact knowledge about the microscopic transitions as encoded by $p_{\mathbf{X}_{t'}|\mathbf{X}_t}$, which is not possible to obtain in most applications.

Finally, Chang *et al.* [4] consider supervenient variables that are "non-trivially informationally closed" (NTIC) to their corresponding microscopic substrate. NTIC is based on a division of $\mathbf{X}_t$ into a subsystem of interest, $\mathbf{X}_t^\alpha$, and its "environment" given by $\mathbf{X}_t^{-\alpha}$. Interestingly, a system being NTIC requires $V_t$ to be supervenient only with respect to $\mathbf{X}_t^\alpha$ (i.e. $V_t = F(\mathbf{X}_t^\alpha)$), as well as information flow from the environment to the feature (i.e. $I(\mathbf{X}_t^{-\alpha}; V_{t'}) > 0$) mediated by the feature itself, so that $\mathbf{X}_t - V_t - V_{t'}$ is a Markov chain. Hence, NTIC requires features that are sufficient statistics for their own dynamics, which is akin to our concept of causal decoupling but focused on the interaction between a macroscopic feature, an agent, and its environment. Extending our framework to agent-environment systems involved in active inference is part of our future work.

### Limitations and open problems

The framework presented in this paper focuses on features from fully observable systems with Markovian dynamics. These assumptions, however, often do not hold when dealing

with experimental data—especially in biological and social systems. As an important extension, future work should investigate the effect of unobserved variables on our measures. This could be done, for example, leveraging Takens' embedding theorem [53–55] or other methods [56].

An interesting feature of our framework is that, although it depends on the choice of PID and ΦID, its practical application via the criteria discussed in Section *Practical criteria for large systems* is agnostic to those choices. However, they incur the cost of a limited sensitivity to detect emergence due to an overestimation of the microscopic redundancy; so they can detect emergence when it is substantial, but might miss it in more subtle cases. Additionally, these criteria are unable to rule out emergence, as they are sufficient but not necessary conditions. Therefore, another avenue of future work should search for improved practical criteria for detecting emergence from data. One interesting line of research is providing scalable approximations for $\mathrm{Syn}_\star^{(k)}$ and $\mathcal{G}_\star^{(k)}$ as introduced in Section *Measuring emergence via synergistic channels*, which could be computed in large systems.

Another open question is how the emergence capacity is affected by changes in the microscopic partition of the system (c.f. Section *Defining causal emergence*). Interesting applications of this includes scenarios where elements of interest have been subject to a mixing process, such as the case of electroencephalography where each electrode detects a mixture of brain sources. Other interesting questions include studying systems with non-zero emergence capacity for all reasonable microscopic partitions, which may correspond to a stronger type of emergence.

## Conclusion

This paper introduces a quantitative definition of causal emergence, which addresses the apparent paradox of supervenient macroscopic features with irreducible causal power using principles of multivariate statistics. We provide a formal, quantitative theory that embodies many of the principles attributed to strong emergence, while being measurable and compatible with the established scientific worldview. Perhaps the most important contribution of this work is to bring the discussion of emergence closer to the realm of quantitative, empirical scientific investigation, complementing the ongoing philosophical inquiries on the subject.

Mathematically, the theory is based on the Partial Information Decomposition (PID) framework [14], and on a recent extension, Integrated Information Decomposition (ΦID) [20]. The theory allows the derivation of sufficiency criteria for the detection of emergence that are scalable, easy to compute from data, and based only on Shannon's mutual information. We illustrated the use of these practical criteria in three case studies, and concluded that: i) particle collisions are an emergent feature in Conway's Game of Life, ii) flock dynamics are an emergent feature of simulated birds; and iii) the representation of motor behaviour in the cortex is emergent from neural activity. Our theory, together with these practical criteria, enables novel data-driven tools for scientifically addressing conjectures about emergence in a wide range of systems of interest.

Our original aim in developing this theory, beyond the contribution to complexity theory, is to help bridge the gap between the mental and the physical, and ultimately understand how mind emerges from matter. This paper provides formal principles to explore the idea that psychological phenomena could emerge from collective neural patterns, and interact with each other dynamically in a causally decoupled fashion—perhaps akin to the "statistical ghosts" mentioned in Section *Causal decoupling*. Put simply: just as particles in the Game of Life have their own collision rules, we wonder if thought patterns could have their own emergent

dynamical laws, operating at a larger scale with respect to their underlying neural substrate (similar ideas have been recently explored by Kent [57]). Importantly, the theory presented in this paper not only provides conceptual tools to frame this conjecture rigorously, but also provides practical tools to test it from data. The exploration of this conjecture is left as an exciting avenue for future research.

## Supporting information

**S1 Appendix. Provides the mathematical proofs of our results, and further details about simulations and preprocessing pipelines.**
(PDF)

## Acknowledgments

The authors thank Martin Biehl, Shamil Chandaria, Acer Chang, and Matthew Crosby for insightful discussions, the creators of the Neurotycho database for opening to the public such a valuable resource, and Yike Guo for supporting this research. F.R. is supported by the Ad Astra Chandaria foundation. P.M. and D.B. are funded by the Wellcome Trust (grant no. 210920/Z/18/Z). A.K.S. and A.B.B. are grateful to the Dr. Mortimer and Theresa Sackler Foundation, which supports the Sackler Centre for Consciousness Science. R.C-H. is supported by Ad Astra Trust, Tim Ferriss, The Nikean Foundation, and The Tamas Family.

## Author Contributions

**Conceptualization:** Fernando E. Rosas, Pedro A. M. Mediano, Henrik J. Jensen, Anil K. Seth.

**Formal analysis:** Fernando E. Rosas, Pedro A. M. Mediano.

**Funding acquisition:** Robin L. Carhart-Harris, Daniel Bor.

**Investigation:** Fernando E. Rosas, Pedro A. M. Mediano.

**Methodology:** Fernando E. Rosas, Pedro A. M. Mediano.

**Supervision:** Henrik J. Jensen, Anil K. Seth, Adam B. Barrett, Robin L. Carhart-Harris, Daniel Bor.

**Writing – original draft:** Fernando E. Rosas, Pedro A. M. Mediano.

**Writing – review & editing:** Henrik J. Jensen, Anil K. Seth, Adam B. Barrett, Robin L. Carhart-Harris, Daniel Bor.

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
