## [Decision Letter · Decision Letter 0]

20 Jul 2020

Dear Dr. Rosas,

Thank you very much for submitting your manuscript "Reconciling emergences: An information-theoretic approach to identify causal emergence in multivariate data" for consideration at PLOS Computational Biology. As with all papers reviewed by the journal, your manuscript was reviewed by members of the editorial board and by several independent reviewers.

The reviewers appreciated the attention to an important topic. Based on the reviews, we are likely to accept this manuscript for publication, providing that you modify the manuscript according to the review recommendations, and to the notes below.

The paper has the cut and the style for a more technical/physical journal, but we see the reasons for your submissions to PLOS CB, and agree that the paper can provide some new technical and conceptual fuel to the debate on the mechanisms of consciousness.

Still, it would be appreciated if, keeping the technical part intact, you could insist a bit more on the biological aspect, and discuss the generalizability to other protocols and datasets.

Furthermore, even if the data are from a public repository, and the calculations are done with a freeware tool, please make sure that you provide on a third party repository some scripts to reproduce the figures. 

Sincerely,

Daniele Marinazzo

Deputy Editor

PLOS Computational Biology

Daniele Marinazzo

Deputy Editor

PLOS Computational Biology

[LINK]

Reviewer's Responses to Questions

**Comments to the Authors:**

Reviewer #1: What can I say - I re-read this piece a few times and I cannot really find any fault. It deals with what, in my mind, is the most challenging piece of mathematical modeling - the modeling of emergence. The authors achieve this in a very accomplished manner both theoretically and- very importantly, practically.

I guess once in a while the only thing left to a reviewer is to congratulate the authors for a masterful submission - and this is the time to do that

Reviewer #2: The authors present a new theoretical framework for thinking about the issue of emergence in complex systems. By considering features extracted from a multivariate dynamical system they define two different types of evolving relationship: causal decoupling and downward causation. They provide specific measure based on PID to quantify these properties, and courser level measures to demonstrate their existence with quantities that are easier to compute in practise.

For a highly technical subject matter the manuscript is very clearly presented and was a pleasure to read. They situate their approach well in comparison to existing techniques from complex systems and the study of consciousness.

Major comments

The examples are very nice but I think ECoG data need a null comparison. There is a strong effect of timescale on the measures considered, but neural signals can be strongly autocorrelated with high power in low frequencies and this autocorrelation can change the bias properties of information theoretic measures as a function of frequency / timescale. I think a full treatment of the bias properties of these measures is certainly beyond the scope of this paper, but a simple shuffled control would be enough to demonstrate whether the profiles seen might be influenced by autocorrelation / filtering parameters. For example, the whole analysis procedure could be repeated with the same filtering and cross-validation pipeline, but at the start all the wrist position timecourses used for training should be permuted (e.g. random circular shift or shuffling across trials). This would lead to a classifier Vt that does not extract any meaningful information from the ECoG, but has the same properties induced by autocorrelation, filtering and regularisation. The plots in panel d) could be shown for 1 such random permutation to see if there is any similar dependence on timescale. (or ideally run many times and mean + spread reported, but understand that may be computationally demanding).

Minor comments

It would be nice to have a bit more discussion and motivation of the definition of supervenient feature, as it seems this is a key area where this framework differs from others (discussed nicely P10-11). In discussion P9 supervenient is described as a property “that can be computed from the state of the system”. But the definition P3 is stronger than that. Would be nice to have some more motivation where the definition is introduced, perhaps with some intuitive examples of features that would be supervenient, and those that would not meet the definition.

P3 “information that is provided independently (and hence redundantly) by both of them”. Not sure I agree that {1}{2} PID term should be called independent information. In the two-bit copy I think its fair to say each predictor provides information about the target independently, but they are not redundant. Maybe could just say “information that is provided by both of them”.

P5 Eqs 5 and 6. Could be clearer for the reader to put the term for D(k) and G(k) directly by the definition, ie “we introduce the downward causation, denoted D(k), and the causal decoupling indices, denoted G(k)” and switch the order of equations to match the order they are presented in the text.

P5. Eq 9. Definition of Un(k) conditioning on multiple terms wasn’t clear to me (ie how to relate to definition on p3)

P8, +/- uncertainty are reported on the numerical results, but not told what these are (s.d., s.e.m., confidence interval etc.) Similarly Fig 5. Error bars not specified.

P9 “linking supervenience to static and causal power to dynamic properties” sentence unclear, add commas or repeat the word properties?

P16: typo “pecifically”.

P17 “calculated using … JIDT”. More details of which method and associated options? (JIDT implements a wide range of estimators)

Reviewer #3: Review of Rosas et al. “Reconciling emergences: An information-theoretic approach to identify causal emergence in multivariate data”

This study proposed possible information theoretic formulations of causal emergence. Specifically, the study distinguished downward causation and causal decoupling and showed critical conditions for the existence of them both theoretically and practically. What is particularly exciting about this study is the application of PID and Phi_ID in this context, because they clarify and disentangle various existing ideas surrounding the notion of emergence. I’m particularly impressed by the theorems presented here, because they allow us to detect causal emergence when we do not know exact ways to construct features. I do not have major concerns regarding the contents of the paper.

I have one general question about causal decoupling. I understand that decoupling exists (i.e. is defined) mathematically in the parity dynamics example. But I wonder whether this can occur in physical interactions which should cover much smaller part of all possible dynamics. So I’m curious to know whether causal decoupling is possible for a dynamics where the dynamics is determined by direct interactions among micro elements (i.e., the state of one element is determined by the states of other elements). Intuitively, if I consider physical implementations of the parity case, I would think we need an additional physical element to directly store the parity of the current state. If this is indeed required, the process itself relies on something like ghost (i.e., not existent, but is used for computation). In other words, what I’m suggesting is that decoupling exists only in mathematics, but not in physics. So I was curious how Gamma behaves in application cases. But the Gamma was not zero, so it may be difficult to find an example of decoupling in real/simulation cases. Could the authors comment on this if they had any thoughts?

For practical applications, the authors focused on k=1 cases. I understand that this is a pragmatic choice, but I wonder how results may different if we consider k>1 cases. Do we need to consider them to be more precise or is there anything that we may miss and be cautious about?

Very minor points:

Page 3: A formal definition of a supervenient feature would help clarify what kind of functions are assumed.

Page 5. The order of eq 5 an eq6 should be reversed for consistency with the text.

Page 11: “Taken’s” should be “Takens’” or “Takens’s” as the name is “Takens”. Relatedly, Satohiro Tajima had a few papers motivated by Takens/Sugihara embedding to compute integrated information using delay embedding.

**Have all data underlying the figures and results presented in the manuscript been provided?**

Reviewer #1: Yes

Reviewer #2: **No: **ECoG data are from a pubilc repository. No reference to availability of simulated data or processed data, and data underlying graphs is not provided in spreadsheet form.

Reviewer #3: Yes

PLOS authors have the option to publish the peer review history of their article (what does this mean?). If published, this will include your full peer review and any attached files.

Reviewer #1: No

Reviewer #2: No

Reviewer #3: No
---

## [Editor Report · Decision Letter 1]

25 Aug 2020

Dear Dr. Rosas,

We are pleased to inform you that your manuscript 'Reconciling emergences: An information-theoretic approach to identify causal emergence in multivariate data' has been provisionally accepted for publication in PLOS Computational Biology.

Best regards,

Daniele Marinazzo

Deputy Editor

PLOS Computational Biology

---

## [Editor Report · Acceptance letter]

4 Dec 2020

PCOMPBIOL-D-20-01042R1 

Reconciling emergences: An information-theoretic approach to identify causal emergence in multivariate data

Dear Dr Rosas,

I am pleased to inform you that your manuscript has been formally accepted for publication in PLOS Computational Biology. Your manuscript is now with our production department and you will be notified of the publication date in due course.

With kind regards,

Nicola Davies
